# Lifetime Prediction Using a Tribology-Aware, Deep Learning-Based Digital Twin of Ball Bearing-Like Tribosystems in Oil and Gas

**Prathamesh S. Desai** *, **Victoria Granja** and **C. Fred Higgs, III** *

Mechanical Engineering, Rice University, 6100 Main St, Houston, TX 77005, USA; victoria.granja@rice.edu
* Correspondence: pdesai@rice.edu (P.S.D.); higgs@rice.edu (C.F.H.III)

**Abstract:** The recent decline in crude oil prices due to global competition and COVID-19-related demand issues has highlighted the need for the efficient operation of an oil and gas plant. One such avenue is accurate predictions about the remaining useful life (RUL) of components used in oil and gas plants. A tribosystem is comprised of the surfaces in relative motion and the lubricant between them. Lubricant oils play a significant role in keeping any tribosystem such as bearings and gears working smoothly over the lifetime of the oil and gas plant. The lubricant oil needs replenishment from time to time to avoid component breakdown due to the increased presence of wear debris and friction between the sliding surfaces of bearings and gears. Traditionally, this oil change is carried out at pre-determined times. This paper explored the possibilities of employing machine learning to predict early failure behavior in sensor-instrumented tribosystems. Specifically, deep learning and tribological data obtained from sensors deployed on the components can provide more accurate predictions about the RUL of the tribosystem. This automated maintenance can improve the overall efficiency of the component. The present study aimed to develop a deep learning-based digital twin for accurately predicting the RUL of a tribosystem comprised of a ball bearing-like test apparatus, a four-ball tester, and lubricant oil. A commercial lubricant used in the offshore oil and gas components was tested for its extreme pressure performance, and its welding load was measured using a four-ball tester. Three accelerated deterioration tests was carried out on the four-ball tester at a load below the welding load. Based on the wear scar measurements obtained from the experimental tests, the RUL data were used to train a multivariate convolutional neural network (CNN). The training accuracy of the model was above 99%, and the testing accuracy was above 95%. This work involved the model-free learning prediction of the remaining useful lifetime of ball bearing-type contacts as a function of key sensor input data (i.e., load, friction, temperature). This model can be deployed for in-field tribological machine elements to trigger automated maintenance without explicitly measuring the wear phenomenon.

**Keywords:** oil and gas; lubricant extreme pressure performance; deep learning; digital twin; automated maintenance; lifetime prediction

## 1. Introduction

The expenditure of fixing an unexpected failure of components in remote, offshore, and subsea oil and gas assets can quickly eat into the profits of fuel recovery companies [1,2]. As opposed to reactive maintenance strategies (i.e., reacting after a failure has happened), pro-active maintenance has been shown to improve the efficiency of subsea oil and gas assets [3,4]. However, it is not easy to quantify all the risks associated with such remote environments [5]. Additionally, a subsea system is a complicated machinery comprised of various systems (viz., manifold, sucker pumps, separators, etc.) that are made up of multiple components such as gears, bearings, and values (refer to Figure 1). Modeling the complicated interdependence of these components and systems is quite challenging.

Understanding tribology or the study of the friction, wear, and lubrication of surfaces in sliding contact can help demystify the physical phenomenon that can cause the failure of components such as gears, bearings, and values [6]. This tribological understanding can result in a more accurate forecast of the maintenance needs of systems in upstream oil and gas [7–9]. Furthermore, recent developments in sensor technology and artificial intelligence provide a novel avenue to build digital twins that can automate maintenance based on analyzing the stream of data output by the components used in such mechanical systems [10–15]. The hierarchy of these digital twins as applicable to an upstream subsea plant is also shown in Figure 1.

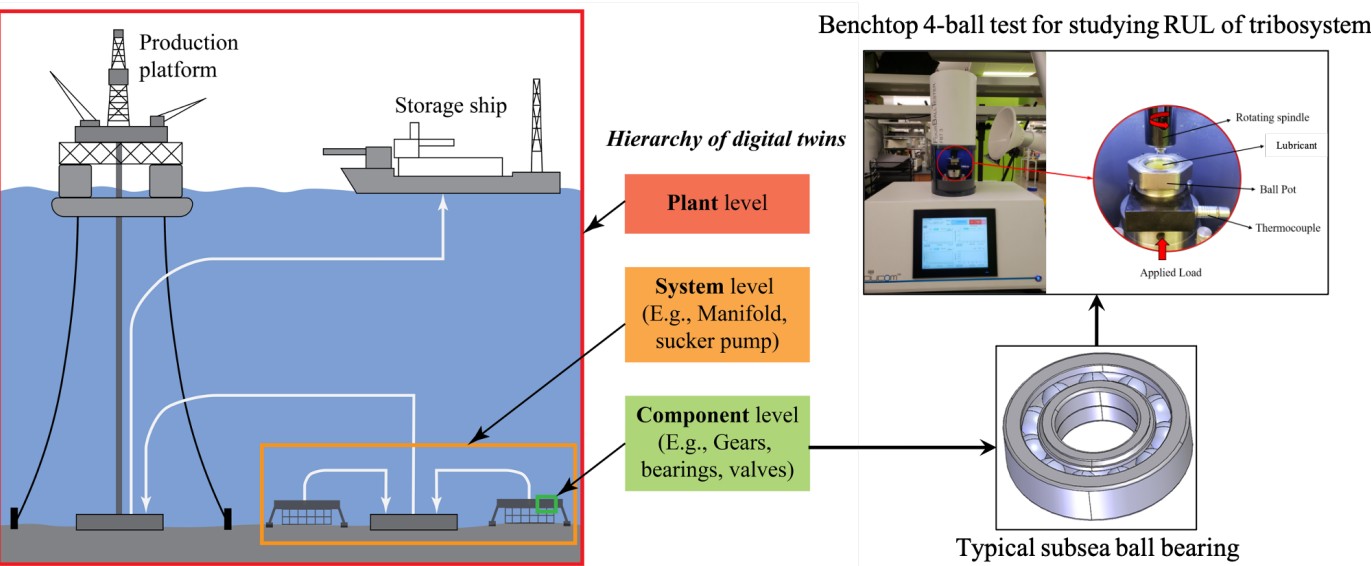

**Figure 1.** (**Left**) Schematic of a typical offshore oil and gas plant highlighting the digital twins. (**Middle**) The associated digital twins and their hierarchy. (**Right**) A typical subsea ball bearing and its benchtop equivalent in the form of a four-ball tester.

### 1.1. Studies on Tribometry in Maintenance

Studies in this category [16,17] have mostly looked at the evolution of the physical properties of the lubricant over the lifetime [18] or the concentration of wear debris particles in the oil over its lifetime [19,20]. The sensors used in such studies are typically far from the tribological interface and measure properties such as water in oil, oil viscosity, and conductivity, but not the quantities such as load, torque, and temperature at or near the interface. The traditional tribological testing of lubricants involves accelerated deterioration of the tribosystem using a pin-on-disk/flat, ball-on-disk/flat, or a four-ball tribometer [6,21,22]. In this paper, a four-ball tribometer was employed, shown in Figure 1.

### 1.2. Studies on Digital Twins in Maintenance

Studies in this category make use of sensor data to train deep learning algorithms such as a fault-predicting super-network [23], an autoencoder (AE) combined with a deep bidirectional gated recurrent unit (bi-GRU) prediction model to predict the RUL [24], and a deep heterogeneous GRU model to forecast tool wear [25]. A few studies using data from physics-based simulations have also attempted to develop a digital twin for maintenance [26–29]. The utility of the latter category of digital twins depends on how accurately the physics-based model can simulate the wear in real systems. As will be shown in Section 3.2, wear in realistic tribosystems is significantly non-linear and poses a severe challenge to physics-based models.

### 1.3. Studies on Machine Learning

Machine learning in general and deep learning in particular have seen a widespread utility from biomedical image segmentation [30] and manufacturing [31], to autonomous vehicles [32] and space [33] applications. A few recent studies have used machine learning as a regression technique in the form of neural networks trained over empirical and theoretical data [34], physics-based model data [35], and physics-plus-empirical data [36]. Time series forecasting is yet another area where machine learning has seen tremendous potential [37]. Researchers have used machine learning for failure prediction on temporal data in optical networks [38], forecasting sales [39], and the energy consumption of buildings [40].

None of those above-mentioned studies drew on the insights of tribological testing to carry out the accelerated deterioration of a tribosystem involving ball bearings and lubricant oil. To the best of our knowledge, this is the first demonstration of the use of interfacial and temporal tribological data and AI, in the form of a multivariate convolutional neural network (CNN), to generate a digital twin (DT) to predict the remaining useful life (RUL) of a tribosystem at the component level. The research methodology of this study is elaborated in Section 2, followed by comprehensive explanations of the tribological experiments and the digital twin modeling framework in Sections 3 and 4, respectively. Experimental and deep learning results are presented and discussed in Section 5. Finally, the conclusions of this study are presented in Section 6.

## 2. Methodology

The research methodology, based on the synergy of experiments and artificial intelligence (AI), used in this study is shown in Figure 2. The experiments involved conducting accelerated deterioration tests on a tribosystem, acquiring the sensor data, and measuring the wear scar. A tribosystem consists of surfaces in sliding contact with intervening fluid and/or particles. This was followed by data processing that involves calculating the remaining useful life (RUL), data cleaning, and splitting into training and test samples for the AI model. The AI-based digital twin (DT) was a deep learning convolutional neural network (CNN) that took the sensor data as inputs to predict the RUL of the tribosystem of interest, here a ball bearing assembly with commercial lubricant oil. Finally, the trained and tested CNN can be deployed on a real component such as a subsea ball bearing (as seen in Figure 1) for prescriptive maintenance. The following two sections elaborate on this research methodology.

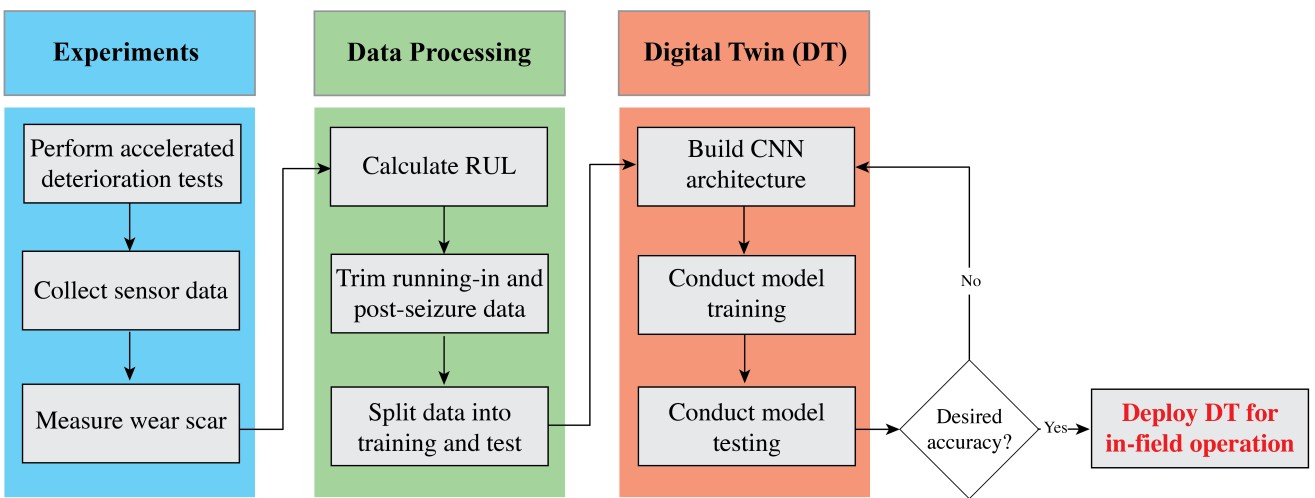

**Figure 2.** Research methodology used in the present study.

## 3. Experiments Using a Four-Ball Tester

The equipment used in this study was a four-ball tester (FBT), in which a steel ball was pressed and rotated against three lower steel balls fully immersed in the lubricant

under test and held stationary in the form of a cradle. This type of tribometer is widely used to determine the wear preventative (WP) and extreme pressure (EP) properties of lubricating oils and greases in sliding steel-on-steel applications. Ball bearings are the most common type of bearing used in mechanical operations, and because of their wide usage, they were selected as the mechanical component of interest for this work. Similar to the four-ball tester, ball bearings operate in sliding motion and have a point contact between the contacting surfaces (refer to Figure 3). The lubricant oil used in this study was a commercial oil developed by Klüber Lubrication called Klübersynth GEM 4 N Series. The oil is developed for off-shore oil and gas applications [41].

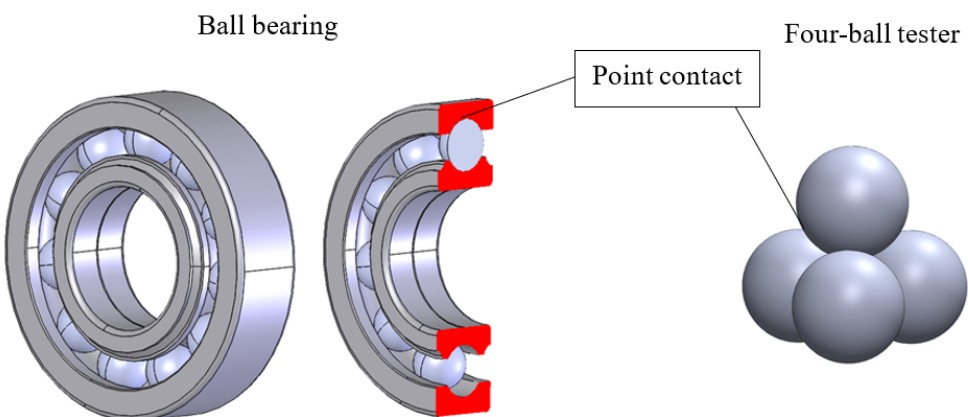

**Figure 3.** (**Left**) Point contact in a real ball bearing. (**Right**) Idealization of a point contact in a four-ball tester.

### 3.1. Extreme Pressure Testing of the Lubricant Oil

First, the extreme pressure properties of the lubricating oil utilized in this study were determined using a four-ball tester. The four-ball tester measurement of the extreme pressure properties of lubricating fluids is described in the ASTM standard D2783 [22]. The purpose of this test was to determine the load-carrying capacity of a lubricating fluid under high load applications such as bearings and gears.

The load, duration, temperature, and rotational speed were set following the ASTM standard mentioned above. A series of tests of 10 s in duration was conducted at standard-defined increasing loads until welding occurred. At the welding point, all four balls were welded at the respective contact points. For each test, the temperature of the lubricant was regulated at 18 °C to 35 °C (65 °F to 95 °F) as the corresponding load was applied. Once these conditions were reached and maintained constant, the upper ball was rotated at 1770 rpm for 10 s. A new set of balls was used for each test at a given load.

Under this methodology, the load-carrying capacity of the test lubricant was found to be 1961 N.

### 3.2. Remaining Useful Life Measurements

Accelerated deterioration tests were then conducted at a load of 1569 N, which was the last non-seizure load of the test lubricant, i.e., load corresponding to one level lower than the welding load of 1961 N. The frictional torque, temperature, load, and speed were documented as a function of time for the duration of the tests. The rotational speed and temperature were the same as those of the extreme pressure tests.

The temporal evolution of the wear scar was determined by conducting interrupted tests. The conditions of the test were the same as those of the accelerated deterioration tests. Here, the test was interrupted every 10 s to measure the wear scar diameter (WSD) of the lower three balls. The original balls and oil were used throughout the test.

The WSD was obtained by measuring the diameters of the scars produced on the three stationary balls after each test interval, as shown in Figure 4. The procedure to measure

the WSD involved first the draining of lubricant oil from the ball-pot. Next, the three balls were left clamped in the ball-pot, and the assembly was placed in the special-purpose microscope. This microscope allowed the measurement of these wear scars located in a plane at an angle with the base. The microscope used was equipped with a calibrated measuring scale that was readable to an accuracy of 0.01 μm. Two measurements were documented for each ball, viz., one along a radial line from the center of the holder and the other along a line 90° from the first measurement. The reported WSD was the arithmetic average of the six measurements (i.e., 2 per ball).

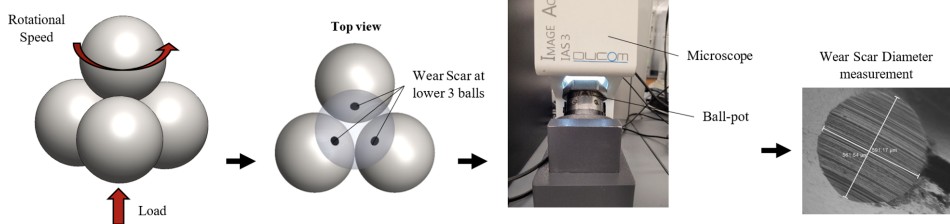

**Figure 4.** Wear scar diameter (WSD) measurement after each test. The special-purpose microscope allows for the measurement of the WSD on the lower three balls without removing the balls from the ball pot. Each reported WSD is the average of the WSD measurements on the lower three balls.

It was revealed that the wear scar remained almost constant for a period before reaching the seizure point, characterized by a sharp increase in frictional torque, after which a sudden increase in the measured scar diameter was evidenced. Figure 5 exemplifies the three distinguished regions after conducting the continuous and interrupted tests. It also includes images of how the wear scar diameter progressed with time.

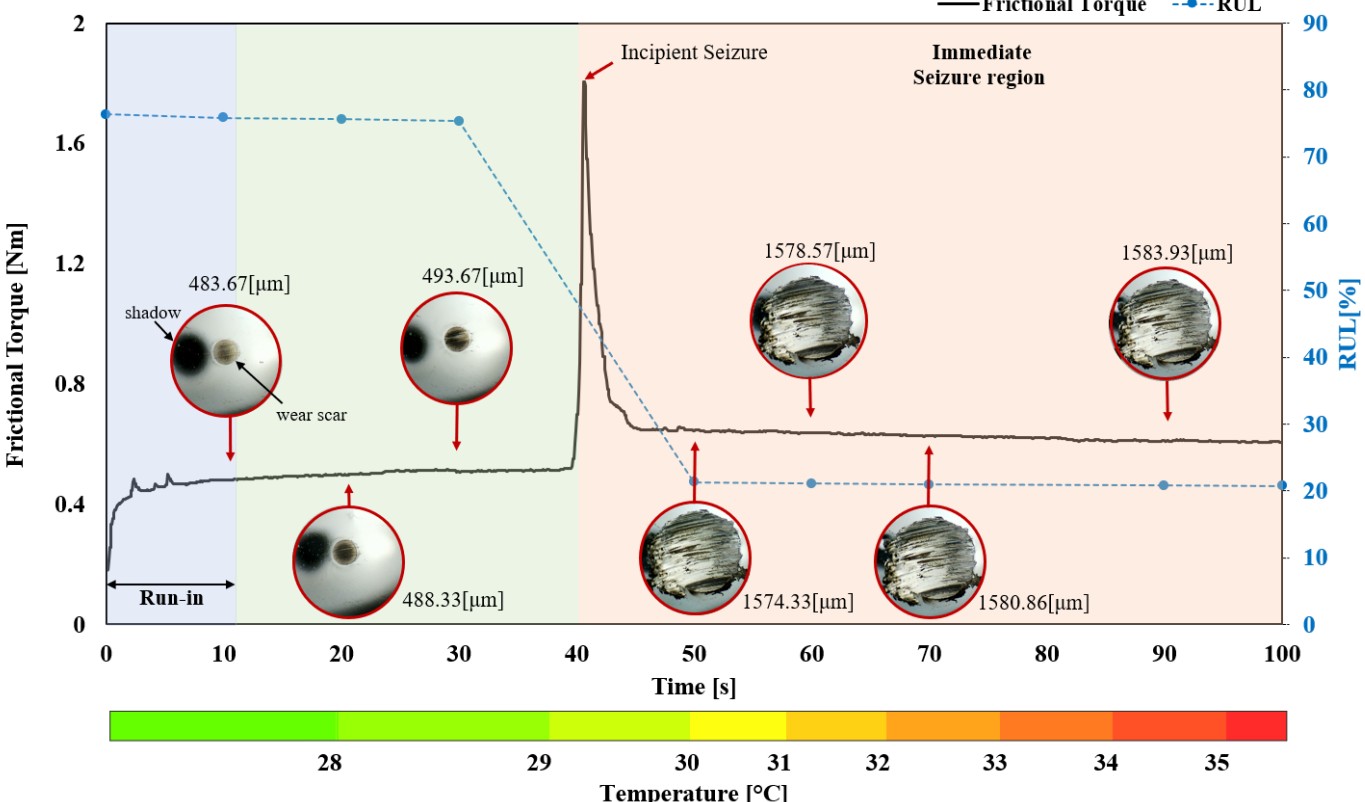

**Figure 5.** Wear scar, temperature, and RUL progression with time in a typical four-ball test. Note that in the first three images, the wear scar is in the center. The dark dot to the left of the wear scar is the shadow cast by the microscope objective.

The *first region*, referred to as the run-in region, represents the initial process that occurs when the balls are brought together under a non-zero normal force and slide relative to one another, causing a plastic deformation where asperities on contacting surfaces conform to each other in a short time [21].

The *second region*, referred to as the proper operation region, represents a steady wear scar and almost uniform frictional torque. It was evidenced that wear scar did not progress linearly with time as one might assume intuitively. Instead, the wear scar diameter for the region of proper operation was found to be almost the same as the Hertzian contact wear scar diameter that can be calculated based on ASTM standard 2783 [22] as follows:

$$D_h = 8.73 * 10^{-2}(P)^{1/3} \tag{1}$$

where:
$D_h$ = Hertz diameter of the contact area in millimeters
$P$ = static applied load in kilograms-force.

The end of the second region was marked by a sudden increase in the frictional torque. The tribosystem then entered the *third region* of operating conditions, referred to as the immediate seizure region (the region characterized by large wear scars and higher friction than in the non-seizure region [22]). Right after the point of incipient seizure (characterized by a momentary breakdown of the lubricating film noted by a sudden increase in the measured scar diameter and a momentary sharp deflection of the friction-measurement device [22]), the wear scar showed a sharp increase in size. The wear scar was also found to remain almost constant with time in this region. It was expected that the end of this region would be marked by an even higher increase in frictional torque and the catastrophic welding of the 4 balls. In this study, we focused on the incipient seizure point instead of the welding point. We could expect to observe the welding point if the samples were tested for a longer period of time. Making an analogy to traffic lights, Regions 1 and 2 could be represented as the green light or "go", Region 3 as the yellow light or "caution", and the welding point as the red light or "stop".

Three continuous accelerated deterioration tests were conducted in this study. The results of these tests are presented in Section 5.

The main goal of predictive maintenance algorithms is to accurately predict the remaining useful life (RUL) of the machine or component of interest. The remaining useful life can be defined as the expected life remaining before the machine or component requires repair or replacement. The lifetime of the machine or component can be defined in terms of any quantity such as distance traveled (miles), fuel consumed (gallons), repetition cycles performed, or time since the start of operation (days). Likewise, time evolution can mean the evolution of a value with any such quantity [42].

In this study, the remaining useful life (RUL) was defined in terms of the wear scar diameter (WSD). It is defined as follows,

$$RUL(\%) = [1 - (WSD_t/WSD_{max})] * 100 \tag{2}$$

where $WSD_t$ is the wear scar diameter at a given time and $WSD_{max}$ is the maximum wear scar diameter or threshold after which the component is considered lost. This value is specified by the user. In this study, it was defined as 2 mm.

Figure 5 also shows how the RUL and temperature progressed as a function of time in a typical test. As shown, the temperature increased linearly as a function of time. On the other hand, the RUL remained almost constant for Regions 1 and 2 and suffered a sharp decline after the seizure point. As described before, the RUL for this study was a function of the WSD; therefore, as the WSD remained almost constant for Regions 1 and 2, the RUL was also almost constant in these two regions. The RUL suffered a sharp decline, after the seizure point, where an abrupt increase in the WSD was generated. The RUL was considered 100% when no load was applied to the system. As soon as the balls came in contact under a specified load, the RUL decreased. At Time 0, the load was already

applied, and the WSD was considered to be that of a Hertzian contact (473.94 um), so the RUL was approximately 76%. After the seizure point, the RUL changed abruptly to approximately 21%.

## 4. Modeling

The model was a data-driven deep learning neural network ($\mathcal{F}$) that mapped sensor data (viz., normal load, frictional torque, and temperature) for a window size $N$ to the RUL at one time step after $N$:

$$RUL_{(N+1)} = \mathcal{F}([x_1, x_2, ..., x_N], [y_1, y_2, ..., y_N], [z_1, z_2, ..., z_N]) \tag{3}$$

Here, $x, y, z$ correspond to the three sensors measuring normal load, frictional torque, and temperature, respectively.

### 4.1. Data-Driven Digital Twin Based on the Convolutional Neural Network

Convolutional neural networks are widely used in computer vision applications; however, these networks also offer a robust framework to study temporal data collected by sensors [26,43]. The significant features in a CNN are convolutions and pooling. These are described below.

#### 4.1.1. Kernel, or Filter, or Feature Detector and Convolutions

As the name suggests, the function of a kernel $\mathcal{K}$ is to extract or filter helpful information from the raw data. The kernel operation involves additive multiplication, and the size of the kernel (a matrix in 2D for images, but a vector for 1D time series) is user-defined. The actual values of the kernel vector are tuned during the network training stage. The non-linearity in the network is introduced by using an activation function $\mathcal{A}$. Typical examples of activation functions are ReLU, sigmoid, and tanh. The entire operation is called convolution. A regular convolution is as seen in Figure 6a. The distance by which a kernel moves or slides across the data is known as the *stride*. The values at the boundaries of the data series are managed using what is known as *padding*. In forecasting a time series, it is important to "not look into the future" to perform a prediction in the past. Thus, a causal convolution, as seen in Figure 6b, with causal padding was used in this study. Mathematically, this can be expressed as:

$$H_j = \mathcal{A}\left[\mathcal{K}[x_{(j-K_s):j}] + b\right] \tag{4}$$

Here, $H_j$ is the $j^{th}$ hidden neuron in the first convolutional layer, $K_s$ is the size of the kernel vector, and $b$ is the bias term.

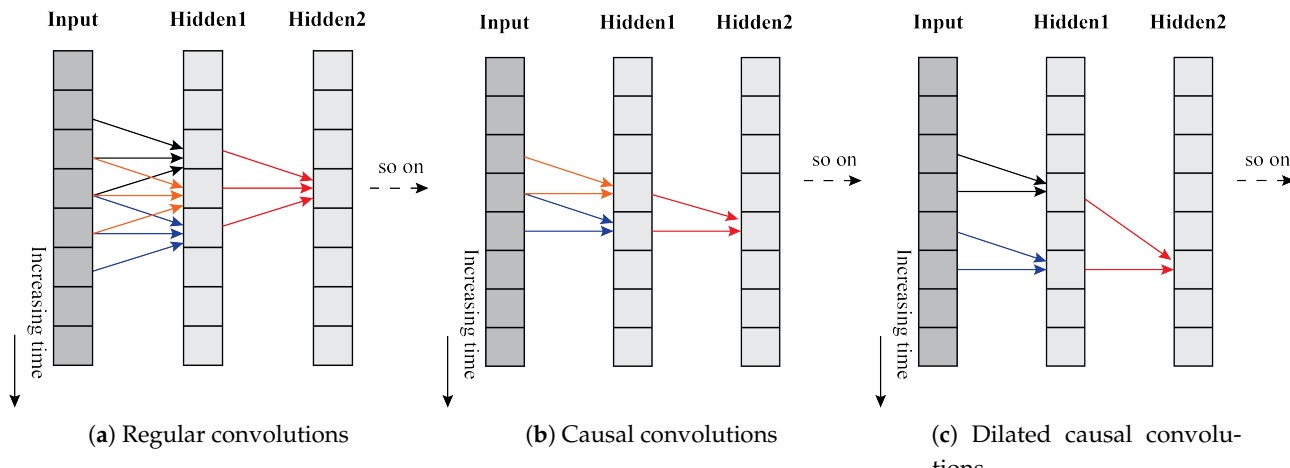

(**a**) Regular convolutions      (**b**) Causal convolutions      (**c**) Dilated causal convolutions

**Figure 6.** Different types of 1D convolutions for a time window of 8 and a single feature or sensor.

### 4.1.2. Dilated Causal Convolutions

A peculiar type of causal convolution, used in Google's Wavenet architecture, is called dilated causal convolution [44] (refer to Figure 6c). In these convolutions, intermediate data values are skipped, and thereby, subsequent layers can see a greater chunk of the time series. In this study, we used dilated causal convolutions with causal padding.

### 4.1.3. Pooling

However, another way to condense the information from the data is to apply a pooling layer after the convolutional layer. A pooling layer either averages the values over a specific kernel size or chooses the maximum value over a kernel size. Pooling is very powerful when dealing with image data as it helps reduce the size of the data, but for time series data, pooling may result in the loss of important information such as a sudden dip in a sensor value [26]. Thus, pooling was not employed in the CNN used in the current study.

### 4.1.4. Dropout

A well-known problem with deep learning neural networks is overfitting. Overfitting refers to a network learning the subtle details in the data and not the trend in the data. An overfitted network has near-perfect accuracy for training samples, but fares poorly for testing samples. There are various ways to avoid overfitting. The simplest of all is to introduce dropout in network training. As the name suggests, dropout drops out a certain number of randomly selected neurons (by making the weights zero) during the training phase. These neurons are turned on during the test phase.

### 4.2. Proposed CNN Architecture

Based on the above concepts, the deep learning neural network used in this study is shown in Figure 7. The input comprised a chunk or a window of sensor data (i.e., normal load, frictional torque, and temperature), and the output was a single RUL prediction as a time point of one beyond the window size. The hidden layers comprised dilated causal convolutional layers, a flatten layer, and a fully connected dense layer for regression. The CNN parameters used in the current study are shown in Table 1, and the resulting CNN architectures are shown in Table 2. The loss function used in this CNN training was Heuber [45], as it is quadratic at lower error values and linear at higher error values. This loss function allows for rapid gradient descent at high errors, but slower gradient descent at low errors, so missing the minimum is less probable. The Adam optimization algorithm [46] was used to perform the gradient descent. The authors of the present study used Google's TensorFlow library [47] to execute this CNN architecture numerically.

**Table 1.** Parametric study with the CNN described in Section 4.

| Parameter | Value |
|---|---|
| Window ($N$) | 16, 32, 64, 128 |
| Number of kernels | 64, 128 |
| Kernel size ($K_s$) | 2 |
| Stride | 1 |
| Dropout | 0.2 |
| Number of neurons in the final dense layer | 128, 256 |
| Activation function for all layers ($\mathcal{A}$) | ReLU |
| Padding | Causal |
| Pooling | None |
| Optimizer | Adam |
| Loss function | Huber |
| Epoch | 500 |

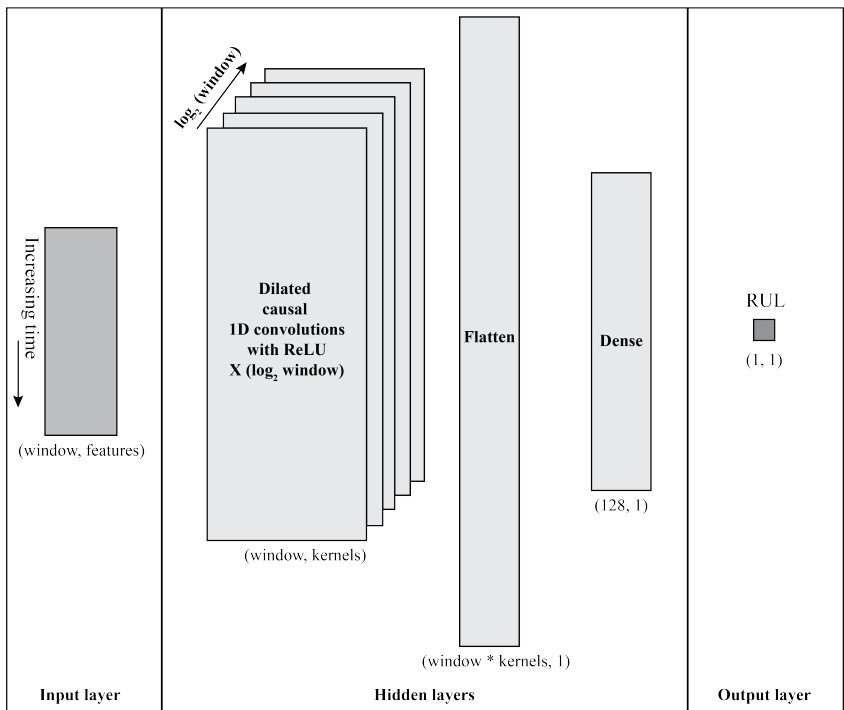

**Figure 7.** The architecture of the deep neural network with dilated casual 1D convolutions used in the current study. Here, features refers to the number of sensors (three in the case of the present study). And, asterisk refers to multiplication.

**Table 2.** Neural network architectures based on the parameters described in Table 1.

| NN Architecture | Window | Filters | Dense Layer Neurons | # of Trainable Params |
|:---:|:---:|:---:|:---:|:---:|
| 1 | 16 | 64 | 128 | 156,545 |
| 2 | 16 | 64 | 256 | 287,873 |
| 3 | 16 | 128 | 128 | 361,985 |
| 4 | 16 | 128 | 256 | 624,385 |
| 5 | 32 | 64 | 128 | 295,873 |
| 6 | 32 | 64 | 256 | 558,273 |
| 7 | 32 | 128 | 128 | 657,025 |
| 8 | 32 | 128 | 256 | 1,181,569 |
| 9 | 64 | 64 | 128 | 566,273 |
| 10 | 64 | 64 | 256 | 1,090,817 |
| 11 | 64 | 128 | 128 | 1,214,209 |
| 12 | 64 | 128 | 256 | 2,263,041 |
| 13 | 128 | 64 | 128 | 1,098,817 |
| 14 | 128 | 64 | 256 | 2,147,649 |
| 15 | 128 | 128 | 128 | 2,295,681 |
| 16 | 128 | 128 | 256 | 4,393,089 |

## 5. Results and Discussion

### 5.1. Processed FBT Experimental Data

In Figure 5, it is explained how the measurements of the WSD, temperature, frictional torque, and RUL behave in a typical test. In this section, each measurement for three different four-ball tester experiments is isolated.

Figure 8 shows the processed data and RUL from the three FBT experimental runs at a normal load of 1569 N and a rotational speed of 1770 rpm. In all three FBT experiments, the normal load (refer to Figure 8a) increased a bit before rapidly decreasing in the post-seizure region (refer to Figure 5). The frictional torque for all three experiments (refer to Figure 8b)

showed the behavior as that seen in Figure 5. The temperature of the tribosystem as measured by the thermocouple (refer to Figure 8c) showed an increasing trend for the entire duration of the experiment. The corresponding RUL curves, based on the Equation (2), for the three FBT experiments, are shown in Figure 8d. The WSD, an average of six measurements (two per ball) on the lower three balls, for all three tests, was measured only once per test at the end of the test, and the WSD in the proper operating region was previously shown to be equal to the Hertz contact diameter, as given by Equation (1). The RUL of 30% was reached at around 30 s, 55 s, and 45 s, respectively, for the three tests. These measurements highlight the physical complexity of tribosystems and the unpredictability of the associated RUL. This data served as the data set for the deep learning model.

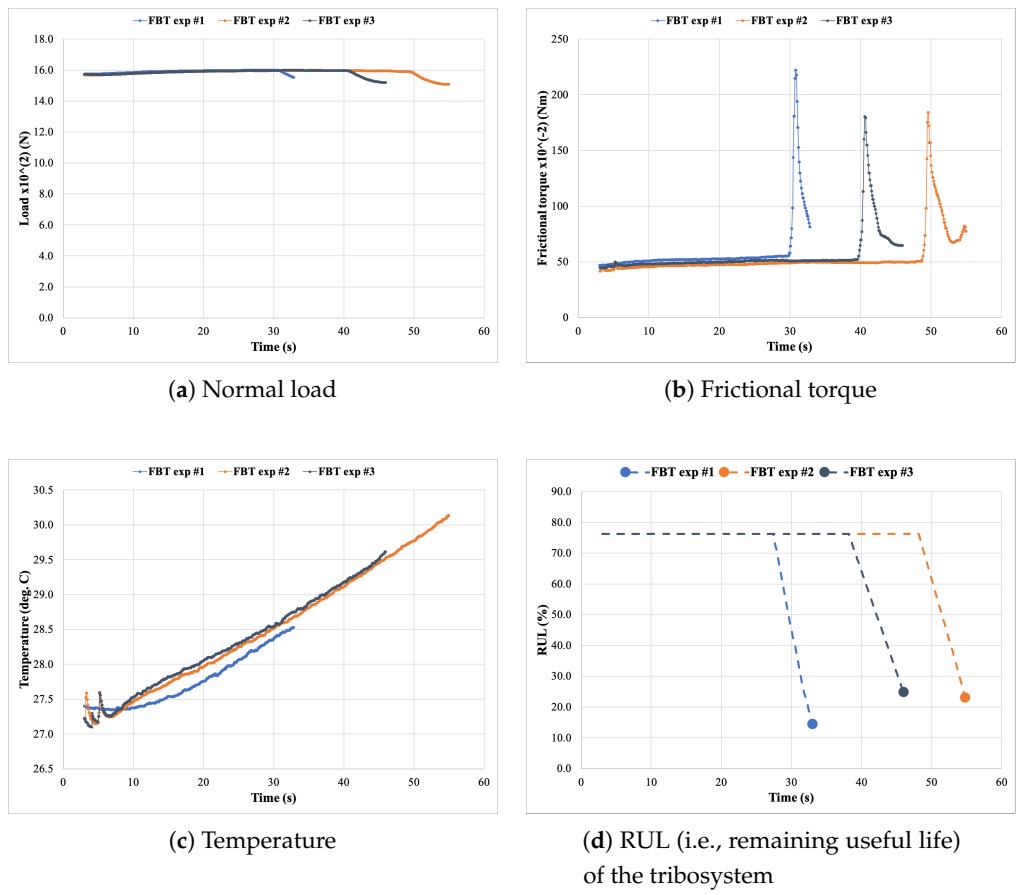

(**a**) Normal load

(**b**) Frictional torque

(**c**) Temperature

(**d**) RUL (i.e., remaining useful life) of the tribosystem

**Figure 8.** Processed data from the three FBT experiments that would serve as the data for the deep learning model.

### 5.2. Digital Twin Model

The various CNN architectures shown in Table 2 were examined for their performance when the data from FBT exp #2 and FBT exp #3 were used for training and those from FBT exp #1 were used for testing. The Pearson correlation coefficient (R) and mean absolute error (MAE) for all the architectures are summarized in Table 3. A value of R close to 100% and that of the MAE close to zero are equivalent to higher accuracy. A general observation about the table is that as the number of trainable parameters (see the last column in Table 2) increases, the training accuracy goes up, and so does the training time. Testing accuracy, however, does not necessarily go up with the increase in the number of trainable parameters. For example, NN Architecture #16 had the maximum number of training variables with a quite low testing accuracy. This lower accuracy highlights the fact that over-parameterization can result in overfitting. Considering the overall table values, NN Architecture #9 was chosen as the winning one as it gave significantly better training and testing accuracy at a lower training time. The results for this NN Architecture #9 are

shown in Figure 9. The difference between the experimental values and model predictions was hardly noticeable for the training samples (i.e., Figure 9b,c). However, the model predictions for the test sample (i.e., Figure 9a) showed quite a bit of mismatch. In particular, the digital twin was slower by 2 s to predict the steep decline in the RUL values when compared to FBT exp #1. Furthermore, the end time RUL predicted by the digital twin was around 40%, a fair bit higher than the FBT exp #1 ground truth of 15%. A point to note is that none of the data from FBT exp #1 were seen by the digital twin.

**Table 3.** Performance of the various NN architectures mentioned in Table 2. FBT Experiments 1 and 2 were used for training and FBT Experiment 3 for testing the NN.

| NN Architecture | Training Time | R Training (%) | MAE Training | R Test (%) | MAE Test |
|---|---|---|---|---|---|
| 1 | 2 m 33 s | 99.6 | 0.666 | 84.4 | 5.86 |
| 2 | 2 m 32 s | 99.65 | 0.885 | 87.4 | 6.41 |
| 3 | 2 m 54 s | 99.72 | 0.854 | 79.13 | 5.97 |
| 4 | 2 m 47 s | 99.72 | 0.654 | 87.2 | 5.48 |
| 5 | 2 m 49 s | 99.8 | 0.784 | 89.1 | 5.69 |
| 6 | 2 m 56 s | 99.8 | 1.25 | 84.7 | 6.91 |
| 7 | 3 m 9 s | 99.7 | 1.11 | 91.9 | 4.13 |
| 8 | 3 m 17 s | 99.7 | 0.371 | 93 | 4.64 |
| **9** | **3 m 18 s** | **99.8** | **0.59** | **95.3** | **4.83** |
| 10 | 3 m 28 s | 99.8 | 0.73 | 95.4 | 5.64 |
| 11 | 4 m 4 s | 99.8 | 0.31 | 90 | 5.1 |
| 12 | 4 m 12 s | 99.8 | 0.26 | 90.5 | 6.04 |
| 13 | 3 m 58 s | 99.8 | 0.64 | 94.4 | 7.35 |
| 14 | 4 m 6 s | 99.8 | 0.26 | 95.9 | 5.86 |
| 15 | 5 m 1 s | 99.9 | 0.39 | 93.8 | 6.45 |
| 16 | 6 m 22 s | 99.8 | 0.35 | 85 | 9.12 |

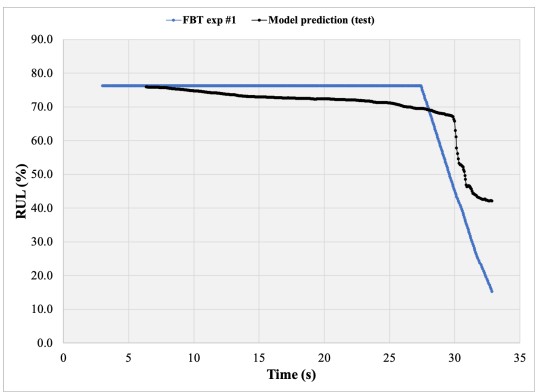

(**a**) Test predictions for FBT exp #1

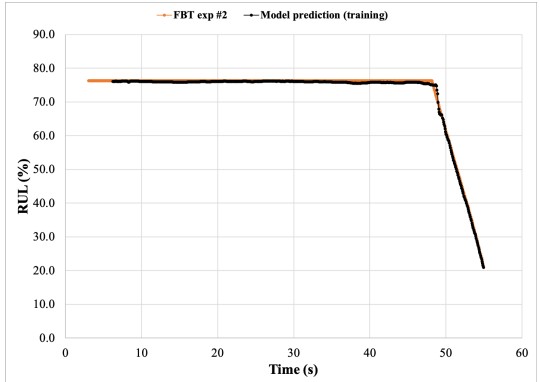

(**b**) Training predictions for FBT exp #2

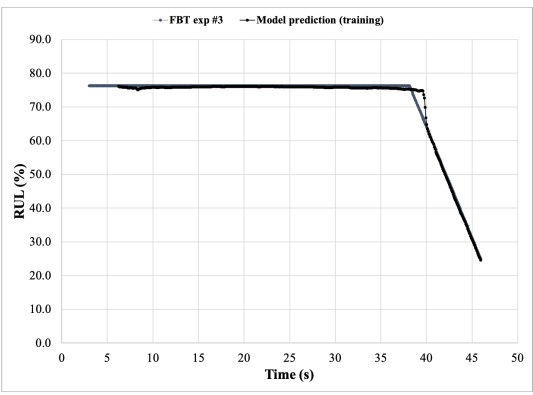

(**c**) Training predictions for FBT exp #3

**Figure 9.** RUL predictions of the deep learning model **trained on FBT Experiments 2 and 3** and tested on Experiment 1. The gray background in (**a**) conveys that the digital twin was tested on this FBT exp.

Due to the expensive and time-consuming nature of these FBT experiments and data processing steps, a permutation of the training and test samples within these three experiments was carried out to check the robustness of the digital twin with NN Architecture #9. A similar data reuse was carried out by [25]. Figures 10 and 11 show the remaining two permutations of the data reuse for the digital twin with NN Architecture #9. Figure 10 shows the predictions of the digital twin when FBT exps #1 and #3 were used for training and FBT exp #2 was used for testing. Here, the digital twin accurately predicted the time of steep decline in the RUL and the RUL at the end of the test. The digital twin slightly over-predicted the RUL values in the initial phase (i.e., proper operation region). Finally, Figure 10 shows the predictions of the digital twin when FBT exps #1 and #3 were used for training and the digital twin was tested against FBT exp #2. Here, the digital twin over-predicted the time of steep decline in the RUL by 1.5 s; however, the RUL at the end of the test was predicted quite accurately. The digital twin perfectly predicted the RUL values in the initial phase (i.e., proper operation region).

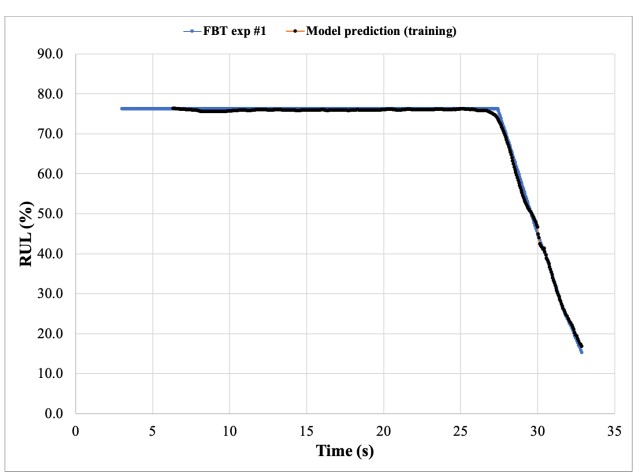

(**a**) Training predictions for FBT exp #1

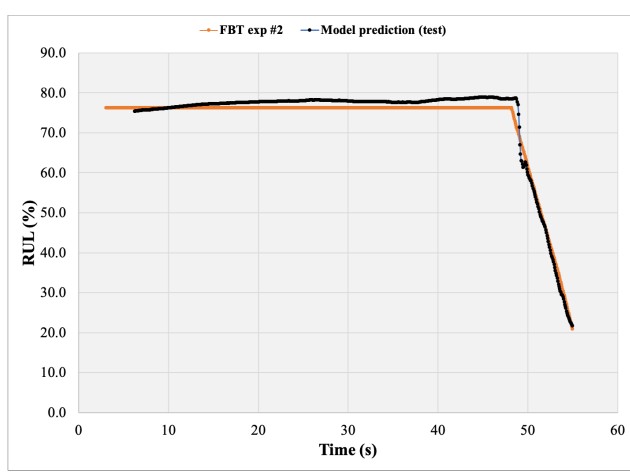

(**b**) Test predictions for FBT exp #2

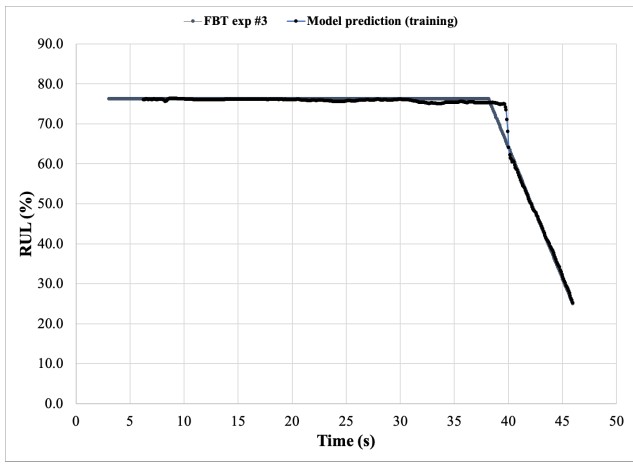

(**c**) Training predictions for FBT exp #3

**Figure 10.** RUL predictions of the deep learning model **trained on FBT Experiments 1 and 3** and tested on FBT Experiment 2. The gray background in (**b**) conveys that the digital twin was tested on this FBT exp.

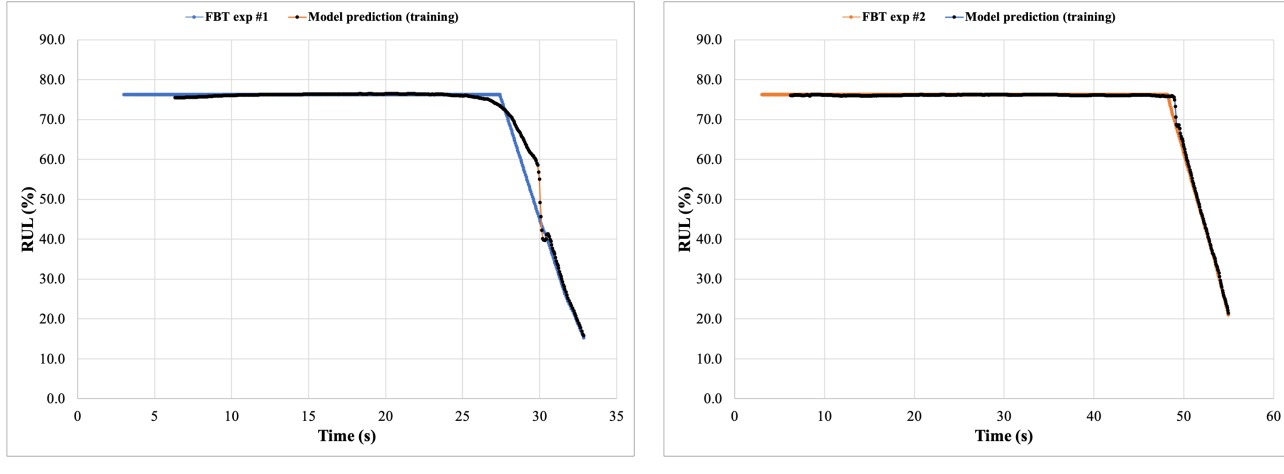

(**a**) Training predictions for FBT exp #1  (**b**) Training predictions for FBT exp #2

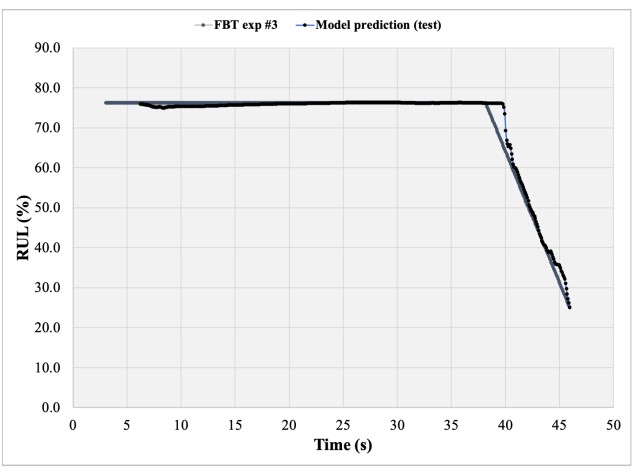

(**c**) Test predictions for FBT exp #3

**Figure 11.** RUL predictions of the deep learning model **trained on FBT Experiments 1 and 2** and tested on FBT Experiment 3. The gray background in (**c**) conveys that the digital twin was tested on this FBT exp.

An improvement in the prediction results can be obtained by training the digital twin on a more significant number of training samples. Nevertheless, the trained digital twin, summarized in Table 4, can now be deployed for in-field subsea applications for real-time prescriptive maintenance decisions. These real-time prognostic decisions would aid in keeping the subsea oil and gas systems cost effective by avoiding superfluous workforce and pre-timed maintenance.

## 6. Conclusions

Automated maintenance of subsea oil and gas plants can significantly improve the plant's efficiency and thereby keep oil and gas producers competitive. In this study, a first of its kind, a tribology-aware and AI-enabled digital twin was proposed for predicting the remaining useful life (RUL) of a tribosystem encountered in upstream oil and gas. The tribosystem was comprised of a ball bearing-like apparatus, a four-ball tester, and commercial lubricant oil for off-shore oil and gas components. The experimental data showed how the operating conditions of a bearing-like tribosystem could suddenly change because of the complex physical interactions of bodies in sliding contact in the presence of a lubricant. These abrupt changes in operating conditions resulted in a sudden drop in the tribosystem's RUL. In tribological applications, one of the ever-present challenges is the inability to measure changes (other than friction) occurring at the interface in situ. Therefore, models

of interfacial parameters are normally developed for this purpose. However, this work involves the prediction of a key interfacial response parameter (i.e., wear scar) as a function of key inputs (e.g., friction) without having an appropriate model. Machine learning opens up the possibilities of predicting tribological performance responses such wear from more tractable sensor data that are difficult to measure in real time. More specifically, this work demonstrated the possibilities of using model-free deep learning techniques to predict the remaining useful lifetime of ball bearing-type contacts as a function of key sensor input data (i.e., load, friction, temperature). The data from three sensors (viz., normal load, frictional torque, and temperature) was used to develop a data-driven digital twin based on a deep learning convolutional neural network (CNN). The CNN was inspired by Google's Wavenet architecture and made use of dilated causal convolutions. A parametric study was carried out to determine the best hyperparameters for the CNN. The training accuracy of the digital twin (i.e., CNN with dilated causal convolutions) was above 99%, and the testing accuracy was above 95%. The digital twin can be subsequently deployed in subsea environments to trigger real-time automated maintenance.

**Table 4.** Final digital twin architecture and performance metrics.

| NN Architecture | Window | Filters | Dense Layer Neurons | # of Trainable Params |
|---|---|---|---|---|
| 9 | 64 | 64 | 128 | 566,273 |

| Training Time | R Training (%) | MAE Training | R Test (%) | MAE Test |
|---|---|---|---|---|
| 3 m 18 s | 99.8 | 0.59 | 95.3 | 4.83 |

**Supplementary Materials:** The following are available online at https://www.mdpi.com/article/10.3390/pr9060922/s1.

**Author Contributions:** Conceptualization, P.S.D.; Data curation, V.G.; Formal analysis, P.S.D. and V.G.; Funding acquisition, C.F.H.III; Investigation, P.S.D., V.G. and C.F.H.III; Methodology, P.S.D.; Project administration, C.F.H.III; Supervision, C.F.H.III; Visualization, P.S.D. and V.G.; Writing—original draft, P.S.D. and V.G.; Writing—review and editing, P.S.D., V.G. and C.F.H.III. All authors have read and agreed to the published version of the manuscript.

**Funding:** This research was partially funded by the Fulbright fellowship and Rice University.

**Institutional Review Board Statement:** Not applicable.

**Informed Consent Statement:** Not applicable.

**Data Availability Statement:** The entire dataset and the trained digital twin are available as supplementary material.

**Acknowledgments:** The authors would like to thank Sunil Garg, founder and CEO of dataVediK, for his industrial insights on data-driven modeling in oil and gas. We also thank Balakumar S. from Ducom Material Characterization Systems for his assistance with maintenance of the four-ball tester.

**Conflicts of Interest:** The authors declare no conflict of interest. The funders had no role in the design of the study; in the collection, analyses, or interpretation of data; in the writing of the manuscript; nor in the decision to publish the results.

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
