# Peer review of "Lifetime Prediction Using a Tribology-Aware, Deep Learning-Based Digital Twin of Ball Bearing-Like Tribosystems in Oil and Gas"

_processes, doi:10.3390/pr9060922_

Round 1

Reviewer 1 Report

The study uses the CNN method on Four-Balls testing tribological behavior to create diagnosing curve based on the experiment. Frictional torque, wear scar diameter on balls, and temperature of oil were obtained by different seizure times (30, 40, and 50 seconds).  The remaining using life is calculated by the ratio of in-situ and final wear scar diameter as shown in Eq.2. It's not reasonable because the wear scar diameter has a large estimation error in the four-balls experiment. Could you provide pieces of evidence to prove the error is acceptable?  Reliability is more important to diagnostic accuracy in the intelligent monitor method development. The authors used two groups of experimental data to train and the other one is used to test as shown in Figs. 9, 10, and 11. Could you provide another experiment, such as seizure time is 80 seconds, to confirm your method?     

Author Response

Our point-by-point responses to the reviewer's queries are included in the attached pdf.

Reviewer 2 Report

This paper presented the lifetime prediction using a tribology-aware, deep learning-based digital twin of ball bearing-like tribosystems in oil and gas. The authors discussed the possibilities of employing machine learning to predict early failure behavior in sensor-instrumented tribosystems. The review comments for this paper are in the following:

  1.  In the title, please use 'and' instead of "&'.
  2.  In Introduction, please compare the different deep learning methods reported in the literature. Please discuss the related works in recent years (Prediction of CO2 absorption by physical solvents using a chemoinformatics-based machine learning model; Comparison of permeability predictions on cemented sandstones with physics-based and machine learning approaches; etc.). Please give the main advances in your method in this work.
  3. In Figure 4, how did you choose the representative WSD measurement?
  4. In Figure 5, are some of the sample figures the same after t=40 s?
  5. Could you please train the predict the data using other literature works besides this experiment?
  6. How could you compare the R2 value with other ML methods in Table 3?
  7. For these influencing factors mentioned in this work, please provide the prediction model for the future data collection. 
  8. A nomenclature is needed to list all the abbreviations in this paper.

Author Response

(The authors gave the same response as above.)

Round 2

Reviewer 1 Report

I have no further comments on the article. The authors have detail replied to my questions. The paper can be accepted as it.

Reviewer 2 Report

Thanks for your revisions.